# Maintenance of Type 2 Response by CXCR6-Deficient ILC2 in Papain-Induced Lung Inflammation

**DOI:** 10.3390/ijms20215493

**Published:** 2019-11-04

**Authors:** Sylvain Meunier, Sylvestre Chea, Damien Garrido, Thibaut Perchet, Maxime Petit, Ana Cumano, Rachel Golub

**Affiliations:** 1Unité de Lymphopoièse, Institut Pasteur, 75015 Paris, France; sylvain.meunier@pasteur.fr (S.M.); sylvestre.chea@gmail.com (S.C.); damiengarrido37@gmail.com (D.G.); thibaut.perchet@pasteur.fr (T.P.); maxime.petit@pasteur.fr (M.P.); ana.cumano@pasteur.fr (A.C.); 2Université Paris Diderot, Sorbonne Paris Cité, 75015 Paris, France

**Keywords:** Immunology, ILC immunity, CXCR6, Lung inflammation

## Abstract

Innate lymphoid cells (ILC) are important players of early immune defenses in situations like lymphoid organogenesis or in case of immune response to inflammation, infection and cancer. Th1 and Th2 antagonism is crucial for the regulation of immune responses, however mechanisms are still unclear for ILC functions. ILC2 and NK cells were reported to be both involved in allergic airway diseases and were shown to be able to interplay in the regulation of the immune response. CXCR6 is a common chemokine receptor expressed by all ILC, and its deficiency affects ILC2 and ILC1/NK cell numbers and functions in lungs in both steady-state and inflammatory conditions. We determined that the absence of a specific ILC2 KLRG1^+^ST2^−^ subset in CXCR6-deficient mice is probably dependent on CXCR6 for its recruitment to the lung under inflammation. We show that despite their decreased numbers, lung CXCR6-deficient ILC2 are even more activated cells producing large amount of type 2 cytokines that could drive eosinophilia. This is strongly associated to the decrease of the lung Th1 response in CXCR6-deficient mice.

## 1. Introduction

Innate lymphoid cells (ILC) are lymphocytes that do not express rearranged antigen receptors such as T and B cells. They have been recently classified into five subsets (NK cells, ILC1, ILC2, ILC3 and LTi cells) based on new data for their development and functions [1]. Members of the ILC family are functionally distinct, with Th cell subset classification similarities such as regulation by soluble factors and specific transcriptional factor profile. Mostly considered as tissue-resident cells, they contribute to the maintenance of tissue homeostasis and the containment of commensals by cytokine production at mucosal barriers. They are also critical during immune responses to pathogens with a concomitant action on tissue inflammation.

NK cells and ILC1 are both Tbet-dependent innate lymphocytes that produce IFNγ whereas ILC2 includes RORα/GATA3-dependent cells that produce type Th2 cytokines. Lung ILC2 play a crucial role in promoting allergic airway inflammation during innate immune responses [2,3], whereas lung NK cells could regulate the early type 2 immune response by an IFNγ retro-control [4]. By the release of IL-5 and IL-13, ILC2 in the lungs respectively mediates eosinophilia and goblet cell activation for the production of mucus [5,6]. Intranasal administration of the protease allergen papain induces the release of type 2 activating cytokines by lung epithelial cells and macrophages to activate ILC2 [7]. Lung explants after papain induction were shown to produce alarmins (IL-25, IL-33) and thymic stromal lymphopoietin (TSLP); all of them can stimulate ILC2 to participate in lung inflammation by secreting Th2 cytokines [7,8]. NK cell functions during lung allergy are not so well understood as studies using different models of lung allergy have been quite contradictory. NK cells can induce the development of asthma in allergen-challenged RAG^−/−^ mice and appear essential for allergic sensitization [9]. On the contrary, NK cells can protect from OVA-induced asthma by contributing to the resolution of allergic lung inflammation in mice. Indeed, they were shown as potent cytotoxic cells against eosinophils and specific CD4^+^ T cells [10]. Lung NK cells can limit viral- and allergen-specific lung type 2 responses through the release of IFNγ [11]. Moreover, in the early stage of papain-induced inflammation, if NK are depleted, the lung inflammation is increased by the enhancement of the ILC2 response [4]. Finally, activated NK cells could restrain the inflammatory effector cells in asthmatic patients [12]. Alternatively, models of contact hypersensitivity have suggested that ILC2 are potent Th1 negative regulators for type 1 immune responses [13].

The bronchial epithelia constitutively express CXCL16, suggesting a role for CXCR6 in lymphocyte homing, recruitment and retention. We have previously shown that ILC precursors both in the adult bone marrow and fetal liver express CXCR6 [14]. We already demonstrated that CXCR6 participates in the retention of ILC precursors into the bone marrow in homeostatic conditions [15]. Most ILC of the intestinal tract express CXCR6 and its deficiency from the ILC3 intestinal subsets perturbs the intestinal ILC homeostasis [16,17]. Homing and cell trafficking use a specific combination of adhesion molecules and chemokine receptors. CXCR6 expression could be correlated to differential homing and/or function of ILC subsets. CXCR6^+^ NK cells have been defined as persistent memory hepatic cells [18]. Most of lung NK cells are circulating, and lineage tracing studies are still needed to confirm the existence of a small percentage of tissue-resident ILC1 in the lungs [19]. In lungs, NK cells have been defined as hypofunctional probably due to their function in maintaining the pulmonary homeostasis [20].

In this study, we took advantage of a mouse model bearing targeted insertion of GFP reporter into the CXCR6 locus (CXCR6^+/GFP^ mice) to follow CXCR6^+^ ILC subsets during lung inflammation. Moreover, it also allowed to inspect the effect of CXCR6 loss on ILC subset functions in lung under steady-state and inflammatory conditions. Competitive reconstitutions demonstrated that for both ILC2 and NK/ILC1, CXCR6 has an intrinsic role on their homing to the lung in steady-state. We used intranasal papain challenges to study ILC2 response and the global type 2 response in absence of CXCR6 expression. We show here that CXCR6 is expressed by a majority of lung ILC2 at homeostasis and that its loss results in a decrease of ILC2 numbers and an exacerbation of the ILC2 activated phenotype. CXCR6 is also present at the surface of ILC1/NK subsets and is important for the ILC1 homing to the lung and IFNγ secretive capacities. While pulmonary ILC2 are reduced in numbers, their capacity to release IL-5/IL-13 is kept by the activated residual subset of ILC2. Finally, in case of lung inflammation, CXCR6 deficiency allowed us to determine that CXCR6 is not crucial to maintain type 2 immune responses. In absence of CXCR6, the increased secretive ILC2 capacities could be related to the decreased capacities of ILC1 to secrete IFNγ. Hence, we suggest that the CXCR6/CXCL16 axis in the lung is mostly related to ILC homing and to Th1 lineage functions as restricting the type 2 response.

## 2. Results

### 2.1. Lung ILC Subsets Differentially Express CXCR6

We examined the proportion of lung ILC subsets that express the chemokine receptor CXCR6 in homeostatic conditions using Rag2^−/−^ CXCR6 reporter mice (Rag2^−/−^ CXCR6^+/GFP^). All our lung preparations were done using an exsanguination protocol that efficiently removed most circulating ILC2, as shown in Appendix A, using control intravital injections of i.v. CD45 antibody before the sacrifice of the mice (Appendix A). The figure proves that exsanguination eliminated most of the circulating ILC2 with only 4% of intravascular ILC2 remaining. On the contrary, NK cells that are known to be all intravascularly positioned remained after exsanguination with a clear positive i.v. CD45 staining suggesting they are actively maintained in this area and not circulating cells.

ILC2 are enriched among lineage-negative lung leukocytes (CD45^+^Lin^−^Nkp46^−^NK1.1^−^). Depending on the expression of KLRG1 and Sca1 markers, three main populations could be separated into KLRG1^−^Sca1^−^, KLRG1^+^Sca1^+^ and KLRG1^−^Sca1^+^ subsets. These subsets were studied for the expression of CXCR6 or GATA3 (Figure 1A). Cells enriched in NK/ILC1 subsets were gated as CD45^+^LinNkp46^+^NK1.1^+^ cells and could serve as a negative control for GATA3 expression (Figure 1A). Considering the positive levels of GATA3 expression, we showed that the KLRG1^+^Sca1^+^ subset was largely constituted of ILC2 whereas the KLRG1^−^Sca1^−^ subset was mostly devoid of them. The KLRG1^−^Sca1^+^ subset was enriched in ILC2 cells. We showed that most lung ILC2 are CXCR6-expressing cells as 85% of the KLRG1^+^Sca1^+^ cells were CXCR6^+^, like 44% of the KLRG1^−^Sca1^+^ subset (Figure 1A). Identical experiments were performed on CXCR6 reporter mice with a wild-type (WT) background (CXCR6^+/GFP^) (Appendix A). The ILC2 KLRG1^+^ subset was mostly CXCR6^+^ in both Rag and WT background (74% and 85%, respectively) whereas the ILC2 KLRG1^−^ subset was less prone to express CXCR6 in a WT background (13% compared to 44% in Rag).

NK cells and ILC1-like cells were selected as CD45^+^Lin^−^Nkp46^+^NK1.1^+^ cells and represent one of the most abundant lymphoid populations of the lung. CXCR6 expression was restricted to a small proportion of these cells (Figure 1B). If we consider that ILC1 exist in the lungs and could be distinguished from NK cells on the basis of the CD127 expression, we determined that NK cells are the most frequent lymphocytes among the lung Nkp46^+^NK1.1^+^ cells. Using this discriminant marker, we showed that CXCR6 was expressed by nearly half of the lung ILC1. On the contrary, only 4–5% of lung NK cells are CXCR6^+^ cells (Figure 1B). The pulmonary NK/ILC1 population still poorly expressed CXCR6 in a WT background with only 8% of the cells being CXCR6^+^ (Appendix A). In conclusion, in homeostatic conditions CXCR6 was expressed by almost all lung KLRG1^+^ ILC2 while it only marked a small population of lung NK/ILC1. However, it is unknown whether CXCR6 is required for maintenance of ILC homeostasis in the lung in steady state and after inflammation.

### 2.2. CXCR6 Deficiency Alters Lung ILC Subset Distribution at Homeostasis

In steady state conditions, the absolute numbers of blood circulating cNK and ILC2 were not significantly modified by CXCR6 deficiency (Figure 2A, left graph). CXCR6-deficient (CXCR6^GFP/GFP^) and -sufficient (CXCR6^+/GFP^) mice were both on the Rag-deficient background. Due to the absence of any T cell contaminants, ILC2 were more easily detected and more frequent. Hence, as already described for ILC2 from Rag-deficient mice, their total numbers in tissues and circulation were increased compared to ILC2 numbers from wild-type mice [21,22]. The digestive tracts of CXCR6-deficient and -sufficient mice were also analyzed for ILC subset respective numbers. As already shown [15,17], ILC3 from the intestinal lamina propria (LP) were decreased in numbers in case of CXCR6 deficiency. Identically, cNK, ILC1, ILC2 and ILC3 subsets from the mesenteric lymph node (mLN) were also decreased in numbers after CXCR6 deletion. Indeed, cNK cells and ILC2 from mLN were decreased by half in CXCR6^GFP/GFP^ mice (Figure 2A, right graph). In lung, the deletion of CXCR6 also reduced ILC absolute numbers by a factor of 3 for ILC2 (Lin^−^CD45^+^NK1.1^−^Sca1^+^) and by a factor of 4 for the type 1 innate lymphocytes (NK/ILC1: Lin^−^CD45^+^NKp46^+^NK11^+^) (Figure 2B).

To have a better understanding of the CXCR6 role for lung ILC, we performed competitive reconstitution experiments by injecting a mix of wild-type (WT) and CXCR6-deficient hematopoietic stem cells (HSCs) (Figure 2C). Sub-lethal irradiated recipient Rag2^−/−^ γc^−/−^ CD45.1 mice were reconstituted with a mix of equal numbers of donor bone marrow-derived Lin^−^Sca1^+^cKit^+^ cells (LSK) from WT CD45.1 mice and CXCR6-sufficient (CD45.2^+^CXCR6^+/GFP^) or CXCR6-deficient (CD45.2^+^CXCR6^GFP/GFP^) mice (Figure 2C). When WT LSK compete with CXCR6-deficient LSK to reconstitute ILC2 and cNK/ILC1 lung populations, lymphoid progenitors from WT origin are significantly favored (Figure 2C,D). No difference was observed in the LSK compartment reconstitution capacities consistent with the absence of CXCR6 expression by LSK. BM CXCR6 expression starts only at the α-lymphoid stage [15]. It indicates that CXCR6 deficiency acts as a cell-intrinsic disadvantage for innate lymphocyte homing to the lungs. Since reconstitution with CXCR6-deficient progenitors is less efficient, we determined here that CXCR6 is important for normal adult NK/ILC1 and ILC2 specific subsets to home to the steady-state lungs (Figure 2D). We cannot clearly determine the cause of this lung homing decrease. It could be due to default of ILC2 recruitment to the lung in absence of CXCR6 or to an upstream default of ILC progenitor maturation in the bone marrow.

### 2.3. In Case of Lung Inflammation, CXCR6 Deficiency Impacts the Number of ILC2 and Their Phenotype

We used intranasal papain protease challenges in order to stimulate a strong allergen airway inflammation [7]. After intranasal papain administration, we observed that the activated ILC2 phenotype was exacerbated in case of CXCR6 deficiency (Figure 3). ILC2 from CXCR6-deficient lungs have increased levels of ST2 expression (Figure 3A,B).

The frequency of lung ILC2 was significantly decreased in CXCR6^GFP/GFP^ mice compared to their CXCR6-sufficient counterparts (Figure 3C, left panel). According to the decrease of their frequency, pulmonary ILC2 were also significantly less numerous in case of CXCR6 deficiency (Figure 3C, right panel), confirming a clear decrease of ILC2 numbers present in inflammatory lungs. In the lung, the inflammatory subset of ILC2 (iILC2) has been described as constituted of ST2^−^ KLRG1^+^ cells in the vascular area and is different from ST2^+^ KLRG1^−^ ILC2 that are defined as the tissue resident subset [23,24]. In CXCR6-deficient lungs, both iILC2 (ST2^−^ ILC2) and nILC2 (ST2^+^ ILC2) subsets were reduced in frequency and numbers even if the strongest effect was observed for iILC2 that were reduced by a factor of 6 (Figure 3D). By comparing ST2^+^ and ST2^−^ subsets’ capacities for secreting the amphiregulin (Areg) in papain-inflamed WT lungs, we found that ST2^−^ ILC2 secreted higher levels of Areg (Figure 3E). Since the ST2^−^ subset was strongly decreased in CXCR6-deficient lungs, we suggest that the wound healing capacities of CXCR6-deficient ILC2 could be questioned (Figure 3E).

Surprisingly, KLRG1 expression was largely increased among nILC2 in CXCR6-deficient lungs and a new KLRG1hi ST2hi subset of ILC2 was defined (Figure 3F). The distribution of these activated nILC2 in the lung was significantly biased in CXCR6-deficient lungs, with 64% of ST2^+^ nILC2 being KLRG1hi compared to 24% in CXCR6-sufficient lungs after papain challenges (Figure 3G). Finally, we determined that in absence of CXCR6, the decrease among nILC2 was due to the specific loss of the KLRG1^−^ subset. The ST2^+^ KLRG1^+^ counterpart was maintained in normal ratio during pulmonary inflammation despite higher levels of ST2 and KLRG1 surface expression (Figure 3H). Identical experiments were performed on CXCR6-deficient mice in a WT background (Appendix A). Globally, the absence of CXCR6 in mice with T and B cells had similar issues with a decrease of lung ILC2 numbers by all subsets (KLRG1^+^, KLRG1^−^, ST2^+^ and ST2^−^) after papain challenges. Moreover, iILC2 still expressed higher levels of KLRG1 in CXCR6-deficient conditions (Appendix A).

### 2.4. CXCR6 Does Not Influence the Capacity of ILC2 to Secrete IL-5 and IL-13 during Inflammation

In order to follow the repercussion of CXCR6 deficiency on ILC2 secretive abilities, we followed IL-5 and IL-13 type 2 immune response after papain lung inflammation (Figure 4). As expected, lung ILC2 subsets were either able to secrete IL-5 alone or combined IL-5/IL-13 after intranasal papain administration (Figure 4A). CXCR6 deficiency did not affect the distribution of these secretive subsets (Figure 4A). Indeed, we measured equal amounts of IL-5 and IL-13 cytokines in the bronchoalveolar lavages (BAL) of CXCR6-sufficient and -deficient mice (Figure 4B). Then, we checked for the resulting levels of eosinophilia in both lungs and BAL. The eosinophil recruitment was similar in CXCR6-sufficient and -deficient mice in coherence with the levels of IL-5/IL-13 (Figure 4C). We demonstrated that after papain stimulation a typical type 2 immune response was mounted in CXCR6-deficient mice. We then decided to look for lung ILC2 expression of other type 2 cytokines such as IL-4 and IL-9 from all genotypes (CXCR6^+/+^, CXCR6^+/GFP^, CXCR6^GFP/GFP^). Their transcript expression levels were quite low (Figure 4D). We observed a tendency to the decrease accordingly with the loss of CXCR6 alleles suggesting that IL-4 and IL-9 expression may be perturbed. However, differences were not significant and mRNA levels were so low in quantity that correlations could not be affirmative.

### 2.5. CXCR6 Deficiency Also Alters the Number and Functions of Type 1 Innate Lymphoid Cells

We decided to analyze pulmonary NK/ILC1 after papain-induced inflammation since some lung NK/ILC1 can express CXCR6. Using CXCR6-sufficient and -deficient animals, we determined the impact of CXCR6 on the repartition of NK/ILC1 at steady-state and after lung inflammation (Figure 5A). In steady-state conditions, KLRG1 was expressed by a very small amount of NK/ILC1 representing between 1–2% of the population and no significant difference was observed (Figure 5A, left panels). However, after papain-induced inflammation, recruited NK/ILC1 were largely CXCR6^+^KLRG1^+^ cells and it was evident that KLRG1 expression was nearly absent from CXCR6-deficient NK/ILC1 of the lungs (Figure 5A, left panels). Moreover, papain administration also increased the frequency of CXCR6^+^ cells in the KLRG1^−^ subset (Figure 5A, left panels). This significant increase was not observed in CXCR6^GFP/GFP^-deficient lungs demonstrating the important role of CXCR6 in the recruitment of these new inflammatory subsets (Figure 5A, left panels). By separating NK/ILC1 based on their CD127 expression, we determined that the CXCR6 deficiency preferentially prevented ILC1-like accumulation in lungs after induction of airway inflammation (Figure 5A, right panels). Indeed, even by diluting CXCR6^+^ ILC1 with CXCR6^−^ ILC1, the decrease in total ILC1 numbers after inflammation was still significant (Figure 5B). On the contrary, NK cells were not significantly different in numbers, mainly because CXCR6^+^ NK cells were too few and their decrease was diffused among the CXCR6 independent population. CXCR6-deficient mice in a WT background were also analyzed for the repartition of ILC1/NK populations after papain stimulation (Appendix A). Globally, the absence of CXCR6 in mice with T and B cells had similar issues with a stronger decrease of lung ILC1 numbers. It is interesting to observe that in presence of T and B cells, NK cells were also importantly decreased after inflammation while their frequency of CXCR6^+^ cells were unchanged at a low 8% rate. It appears that the recruitment of NK cells in case of lung inflammation might be indirectly dependent on CXCR6 expression by lymphocytes.

Finally, we compared IFNγ secretive abilities between CXCR6-deficient and -sufficient mice and observed that IFNγ secreting ILC1 were significantly decreased (Figure 5C). In case of low IFNγ secretion levels, secreting ILC1 only tend to decrease. The dilution of CXCR6^+^ NK cells was again too strong to reveal a significant decrease of IFNγ secretion but a tendency to the decrease was emerging (Figure 5C).

## 3. Discussion

In this study, we assessed the effect of CXCR6 deficiency on lung ILC2 and NK/ILC1 in steady state and after papain-driven inflammation into the lungs. The effect of CXCR6 loss was analyzed using the Rag2^−/−^ CXCR6^GFP/GFP^ mouse model since the GFP reporter insertion simultaneously inactivates the corresponding *Cxcr6* allele [25]. The absence of Rag2 warrants a proper ILC analysis devoid of T-cell contamination. It was shown that tissue ILC2 and ILC3 are elevated in numbers and also secrete increased amounts of cytokine in the Rag-deficient background [21,22]. However, CXCR6-sufficient and -deficient mice are all on the Rag-deficient background to fairly conclude on the CXCR6 role. Moreover, it was previously proven that activated ILC2 induce pulmonary eosinophilia independently of the adaptive immune system [7]. In parallel, we showed that in a WT background CXCR6 is still highly expressed by the subset of KLRG1^+^ iILC2 and by less than 10% of the pulmonary ILC1/NK cells. Hence, we concluded that the main difference found is that CXCR6 is less frequently expressed by the resident lung KLRG1^−^ nILC2 in case of RAG deficiency.

We observed that CXCR6 is an important chemokine that could regulate the type 1/type 2 balanced immune response in the lungs. In homeostatic conditions, we showed that both ILC2 and NK/ILC1 pulmonary subsets are perturbed in their lung homing capacities in case of CXCR6 deficiency. Even if ILC1 were not yet clearly defined in the lungs, we considered the CD127^+^ population of NKp46^+^NK1.1^+^ lung cells as being the lung resident ILC1 population. We also checked that this population was Eomes^−^ contrary to bona fide NK cells.

Competitive reconstitution experiments proved that the CXCR6 cell’s intrinsic role is to regulate ILC homing to the lung. Since we have previously shown that CXCR6-deficient ILC progenitors have a retention problem in the BM, it is equally possible that CXCR6 deficiency induces an upstream lesion in ILC progenitor maturation that consequently limits lung ILC engraftment. Indeed, we also observed a significant decrease of ILC1 and ILC2 in steady state lungs of CXCR6-deficient mice. Competitive reconstitution differences were not highly different (*p* = 0.028) for two main reasons. The first one is coming from the fact that ILC colonize tissue by waves and the bone marrow-derived wave only concerns a fraction of adult tissue ILC2 [26]. The second one is due to the sparse expression of CXCR6 among lung NK/ILC1 reducing the impact of CXCR6 loss. Especially for pulmonary NK cells, only few cells express CXCR6 and the role and effects of this molecule are largely diluted in case of depletion. On the contrary, as CXCR6 is expressed by half of ILC1, the results were significant when ILC1 were separated from NK cells for analyses.

There is no clear explanation of the NK/ILC1 roles in lung allergy. Lung NK cells are described as a circulating hypofunctional subset, with more mature phenotype than in other tissues, that probably originate from the bone marrow [20,27]. KLRG1 is an inhibitory receptor and marker of terminal maturation acquired by mature NK cells that migrate into secondary lymphoid organs [28,29]. Mostly associated to decreasing ability to secrete IFNγ [29], KLRG1 expression on NK cells could also reveal a recent activation and punctual capacity to produce more IFNγ in case of inflammation [30].

In case of lung inflammation such as papain-induced airway hyperresponsiveness, the trafficking of leukocytes is important to recruit inflammatory subsets and we demonstrated that CXCR6 deficiency was preventing the recruitment of new incoming mature activated KLRG1^+^ NK cells or iILC2 [23]. While, in CXCR6-deficient lungs, KLRG1 expression is completely absent from ILC1/NK subsets, the situation is more complex for ILC2 where new KLRG1^+^ ST2^−^ iILC2 are less able to reach the lung while nILC2 become increasingly activated and gain high expression for both ST2 and KLRG1 surface markers. Lung KLRG1^+^ iILC2 were largely enriched in CXCR6-expressing cells confirming that the iILC2 subset trafficking to the lungs is already CXCR6^+^. CXCR6 is important since its abrogation largely decreases the amount of recruited iILC2 and consequently challenges the number of nILC2 by reducing the number of cells able to differentiate into nILC2 [23,24]. Hence, it seems that circulating inflammatory subsets are mostly CXCR6-dependent. The CXCR6^+^ resident subsets probably result from an in situ differentiation of iILC2 that might receive different signals in CXCR6-deficient conditions perturbing their activation state. Indeed, KLRG1^+^ ILC2 accumulation in CXCR6-deficient mice may reveal this inability of ILC2 to be correctly localized in the lung and to further differentiate into ST2^+^KLRG1^+^ and then ST2^−^ nILC2.

We do not know at which stage CXCR6 is crucial; it could be crucial from an early developmental BM stage for the maturation of ILC progenitors as already stated [15] with a direct consequence in CXCR6^+^ intestinal ILC2 subsets. We analyzed the intestinal lamina propria ILC2 and showed that more than half were CXCR6^+^ in steady-state conditions (Appendix A). Moreover, we showed that in CXCR6-deficient mice, ILC2 from the mLN are significantly decreased in numbers. It appears consistent that in CXCR6-deficient mice less iILC2 are available to reach the lung in inflammatory conditions. Hence, CXCR6 may have different roles at different stages of their development resulting in less ILC2 in the lungs with an abnormal localization. However, iILC2 are described as strong producers of cytokines [23,24]. Since CXCR6^+^ cells are mostly pulmonary iILC2, they could be attributed to the pool of cytokine producers. This CXCR6^+^ subset is reminiscent of memory ILC2 [31].

As CXCR6 may have an indirect role in the homing of ILC2 to the lungs, other chemokines have been shown as important for the homing to the lungs. CCR4, CCR8 and CCR1 are expressed by ILC2 in steady-state conditions and are implicated in lung ILC homeostasis with a decrease of ILC2 numbers in case of CCR4 deficiency [32,33]. After IL-33-induced inflammation, CCR8 and CCR1 are increased at the surface of ILC2 and blockade of CCR8 could reduce both ILC2 motility and secretion suggesting an important role for their homing to the lungs in the inflammatory context [33]. Moreover, the expression of CCR8 has been correlated with the expression of its CCL8 ligand by inflamed epithelial cells [33]. After the induction of inflammation, lung ILC2 also express CCL1, another CCR8 ligand, and this expression enhances their motility, proliferation and survival when interacting with the inflamed epithelium [32].

We observed a clear decrease of IFNγ secretion activities by CXCR6-deficient lung ILC1. Despite the absence of iILC2 in CXCR6-deficient mice, levels of type 2 cytokines were identically maintained and the ratio of recruited eosinophils was consistently similar. It was previously shown that alarmins expand the lung nILC2 and their IL-13 secretion. Moreover, lung nILC2 could be negatively regulated by IFNγ [4,34]. We suspect that CXCR6-deficient nILC2 harboring high levels of ST2 and KLRG1 in the lungs were overactivated due to an absence of negative regulators. Hence, these lung nILC2 are no longer suppressed by IFNγ but are overactivated by IL-25 and IL-33 after papain induction. In CXCR6-deficient lungs that contain less ILC2, we suggest that type 2 maintenance is explained by the decrease of the type 1 immune response.

IL-9 was shown to be produced by iILC2 in papain-induced inflammation after a combined IL-33/IL-2 stimulation. In our Rag models, IL-2 levels are probably more limited, explaining why IL-9 transcripts are lowly detected. Nevertheless, lung ILC2 cannot induce normal levels of IL-9 and IL-4 secretion in CXCR6 deficiency. Other models of inflammation should be tested to evaluate the clear IL-4 and IL-9 secretive state of ILC2 in CXCR6-deficient conditions. We showed that ST2^−^ iILC2 are responsible for the secretion of an important amount of amphiregulin in the case of papain-induced lung inflammation. ILC2 in response to IL-33 and in situations of tissue damage secrete the amphiregulin as part of the type 2 cellular response [35,36]. Due to the absence of this ILC2 subset, the capacity of lung epithelium to return to homeostasis in CXCR6-deficient mice is probably challenged.

CXCR6-deficient mice in a WT background were also analyzed for the repartition of ILC1/NK populations after papain stimulation. Globally, the absence of CXCR6 in mice with T and B cells had similar issues with a stronger decrease of lung ILC1 numbers. It is interesting to observe that in presence of T and B cells, NK cells are also importantly decreased after inflammation while their frequency of CXCR6^+^ cells are unchanged and maintained at a low rate (8%). It appears that the recruitment of NK cells in case of lung inflammation might be indirectly dependent on CXCR6 expression by T or B lymphocytes.

In conclusion, we showed here that while CXCR6 is important in the first homeostatic event of lung homing, this chemokine is also important in case of lung induced inflammation as it could block the recruitment of inflammatory subsets of both type 1 and type 2 immunity. We demonstrate that CXCR6 is intrinsically related to IFNγ secretion abilities, explaining why its deficiency favors type 2 immune responses. Then, CXCR6 deficiency could be detrimental in case of asthma inflammation where it sustains type 2 response by the decrease of restricting type 1 secretion but limits the possible epithelial wound healing.

## 4. Materials and Methods

### 4.1. Mice and Animal Facilities

Rag2^−/−^ CXCR6^+/+^, Rag2^−/−^ CXCR6^GFP/+^, Rag2^−/−^ CXCR6^GFP/GFP^, CXCR6^GFP/+^, CXCR6^GFP/GFP^ and CD45.1 Rag2^−/−^ γc^−/−^ mice were bred in the animal facilities at Pasteur Institute, Paris. Mice were cared for in accordance with Pasteur Institute guidelines in compliance with European animal welfare regulations; all animal studies were approved by Pasteur Institute Safety Committee according to the ethic charter approved by French Agriculture ministry and to the European Parliament Directive 2010/63/EU. Project 02080.02 recorded the 5/10/2016.

### 4.2. Inflammation of Lung

Induced allergic lung inflammation in mice was generated by papain administration. A total of 40 µg of papain (Sigma Chemical, St. Louis, MO, USA) was administered intranasally to anesthetized mice by two injections (J0 and J4), 20 µg for each injection in 50 µL PBS 1X. ILC2s and eosinophils were analyzed 5 days after the first administration.

Anesthesia was a mix of Ketamin (Merial, Lyon, France, 100 mg/kg) and Xylazin (Bayer, Leverkusen, Germany, 10 mg/kg).

### 4.3. Cell Preparation

Lungs were harvested and washed by PBS 1X injection. After being cut in small fragments, lungs were incubated 45 min at 37 °C and 5% CO_2_ in RPMI medium (Gibco ThermoFisher, Waltham, Mass., USA) plus collagenase type IV (Gibco ThermoFisher, Waltham, MA, USA, 275,000 UI/mg) and DNAse I (Roche, Basel, Switzerland, 50 µg/mL). Lungs were mechanically crushed, spun and resuspended in a 40% solution of Percoll (GEHealthcare, Nightingales, UK). After centrifugation for 20 min at 600× *g*, cells were collected in cold HBSS medium (Gibco ThermoFisher, Waltham, MA, USA) plus 1% fetal calf serum (FCS).

Small intestines were harvested and washed of their contents by PBS 1X; Peyer’s patches, if present, were removed. After being cut open longitudinally, small intestines were cut in 1-cm fragments. Fragments of small intestine were incubated 30 min at 37 °C and 5% CO_2_ in RPMI medium plus 20% FCS and HEPES buffer (Sigma Chemical, St. Louis, MO, USA, 10 µM). Small intestines were vortexed thoroughly for 4 min for removal of epithelial cells and intraepithelial lymphocytes. Remaining fragments of small intestine were incubated 30 min at 37 °C and 5% CO_2_ in RPMI medium plus 20% FCS, HEPES Buffer and collagenase type VIII (Sigma Chemical, St. Louis, MO, USA, 250 µg/mL) for the isolation of LP lymphocytes. Cell suspensions were spun and resuspended in a 40% solution of Percoll. After centrifugation for 20 min at 600× *g*, cells were collected in cold HBSS medium plus 1% FCS.

Blood was harvested and diluted two times in PBS 1X. Cell blood suspension was put on FICOLL medium (2 volumes of cell suspension for 1 volume of FICOLL). After centrifugation for 20 min at 600× *g*, cells were located in ring between FICOLL and plasma, and were collected in cold HBSS medium plus 1% FCS.

Mesenteric lymph nodes (mLN) were harvested and mechanically dissociated in HBSS medium plus 1% FCS to obtain cell suspension.

All cell suspensions were spun 5 min at 450× *g*; pellets were resuspended in appropriate volume. All cell suspensions were counted on Malassez cell and dead cells were excluded using Trypan Blue.

### 4.4. Lymphocyte Activation

Lymphocytes were activated by culture in OPTIMEM medium plus 10% FCS, β-Mercapto-ethanol (Gibco ThermoFisher, Waltham, MA, USA, 500 µM), penicillin (Gibco ThermoFisher, Waltham, MA, USA, 5 U/mL), streptomycin (Gibco ThermoFisher, Waltham, Mass, USA, 5 µg/mL), Monensin (ThermoFisher, Waltham, MA, USA, 2 µM), Brefeldin A (BD Biosciences, Franklin Lakes, NJ, USA, 3 µg/mL), Phorbol 12-myristate 13-acetate (PMA) (Sigma Chemical, St. Louis, MO, USA, 0.05 µg/mL) and Ionomicin (Sigma Chemical, St. Louis, MO, USA, 1 µg/mL). Cells were cultured for 3 h at 37 °C and 5% CO_2_ and then harvested and stained for analysis.

### 4.5. Flow Cytometry

Streptavidin was coupled to PE-Cy5. The following antibodies were either biotinylated or coupled to fluorochromes (FITC or GFP, PerCP-Cy5.5, PE, PE-Cy7, APC, APC-Cy7, BV421, BV605, BV650, BV711, BV786): Ly76 (TER-119), Gr-1 (RB6-8C5), CD11c (HL3), CD3ε (145-2C11), CD19 (6D5), CD8α (53-6.7), CD5 (53-7.3), TCRβ (H57-597), TCRγδ (GL-3), NK1.1 (PK136), NKp46 (29A1.4), IL-7Rβ (A7R34), Sca-1 (D7), ST2 (DIH9), CD4 (GK1.5), CD45.2 (104), Thy1.2 (53-2.1), ICOS (C398.4A), Gata3 (L50-823), F4/80 (BM8), Siglec-F (E50-2440), IFNγ (XMG1.2), IL-5 (TRFK5) and IL-13 (eBio13A). Lineage cocktails were as follows: CD3ε, CD5, CD8, CD19, F4/80, TCRβ, TCRγδ and Ter119. All antibodies were purchased from BD Biosciences (Franklin Lakes, NJ, USA), eBiosciences (Thermo Fisher, Waltham, MA, USA) or Biolegend (San Diego, CA, USA).

After antibody staining, cells were washed. Cells were stained intra-cellularly using Foxp3 Permeabilzation/Fixation Kit according to manufacturer’s notice (eBiosciences, ThermoFisher, Waltham, MA, USA). Cells were washed prior to cell acquisition. LSR Fortessa (BD Biosciences, Franklin Lakes, NJ, USA) was used for flow cytometry acquisition, with Diva6 software (BD Biosciences, Franklin Lakes, NJ, USA), and analyzed with FlowJo 10 software (FlowJo LLC, Ashland, OR, USA). For visual purposes, only 8000 events maximum are shown in each FACS dot plot. Cells were purified with a FACSAria III sorter (BD Biosciences, Franklin Lakes, NJ, USA). Cells were recovered in PBS for cell injection, or OPTIMEM (Gibco, Thermo Fisher, Waltham, MA, USA) plus 10% FCS for cell culture.

### 4.6. In Vivo Reconstitution

CD45.1^+^ Rag2^−/−^ γc^−/−^ mice were sub-lethally irradiated (400 rad) at least 4 h prior to cell injection. A total of 4000 CD45.1^+^ LSK (Lin^−^ IL-7Rα^−^ c-Kit^+^ Sca1^+^) cells from wild type adult bone morrow and 4000 CD45.2^+^ CXCR6^+^ LSK cells from CXCR6^+/GFP^ or CXCR6^GFP/GFP^ adult bone morrow were sorted and then mixed at a 1:1 ratio for injection. Retro-orbital i.v. injections were performed, and mice were analyzed 8 weeks after injection.

### 4.7. RT-qPCR Analysis

ILC2 were sorted in Buffer RLT (Qiagen, Venlo, Netherlands) containing 2-mercaptoethanol (Sigma Chemical, St. Louis, MO, USA) and were frozen at −80 °C. RNA was obtained with an RNeasy Micro Kit (Qiagen, Venlo, Netherlands) and complementary DNA (cDNA) was obtained with the PrimeScript RT Reagent Kit (Takara, Kyoto, Japan). A 7300 Real-Time PCR System (Applied Biosystems, ThermoFisher, Waltham, MA, USA) and TaqMan technology (Applied Biosystems, ThermoFisher, Waltham, MA, USA) or SYBR Green Technology (Qiagen, Venlo, Netherlands) were used for qRT-PCR analysis. The following primers were from ThermoFisher (Waltham, MA, USA): IL-4 (Mm00445259_m1) and IL-9 (Mm00434305_m1).

### 4.8. Exsanguination Experiment

Anti-CD45 antibody coupled with a fluorochrome was administered to mice by i.v. injection. After 2 min, the mice were sacrificed, organs were treated normally, and cells were labeled as previously described.

### 4.9. Statistical Analysis

All data were submitted to Student’s unpaired bilateral *t*-test. Data were deemed significantly different when * *p* < 0.05, ** *p* < 0.01, or *** *p* < 0.005.

In all figures, histograms are represented as means including SD based on number of replicates indicated in figure legend.

## Figures and Tables

**Figure 1 ijms-20-05493-f001:**
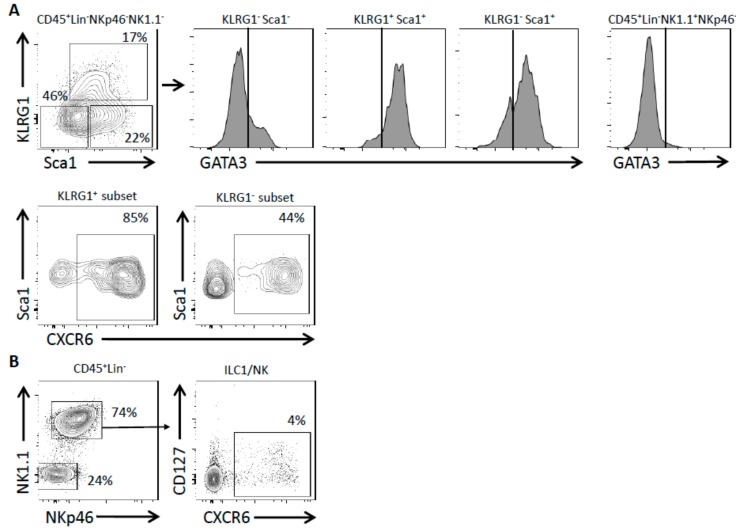
CXCR6 is expressed by lung ILCs. (**A**) Flow cytometry analyses of CD45^+^ Lin^−^ (CD3, CD5, CD19, TCRβ, TCRγδ, Gr1, Ter119, CD8α and F4/80) NKp46^−^ NK1.1^−^ lung cells from Rag^−/−^ CXCR6^+/GFP^ mice. Different subsets are distinguished depending on the expression of KLRG1, Sca1 and GATA3 (GATA3 negative control was performed on CD45^+^ Lin^−^ NK1.1^+^ NKp46^+^ cells). Sca1 and CXCR6 expression from KLRG1^+^ and KLRG1^−^ populations (graphs are representative of *n* = 6). (**B**) Flow cytometry analyses of CD45^+^ Lin^−^ lung cells from Rag^−/−^ CXCR6^+/GFP^ mice. CD127 and CXCR6 expression among ILC1/NK (NK1.1^+^ NKp46^+^) cells (graphs are representative of *n* = 6).

**Figure 2 ijms-20-05493-f002:**
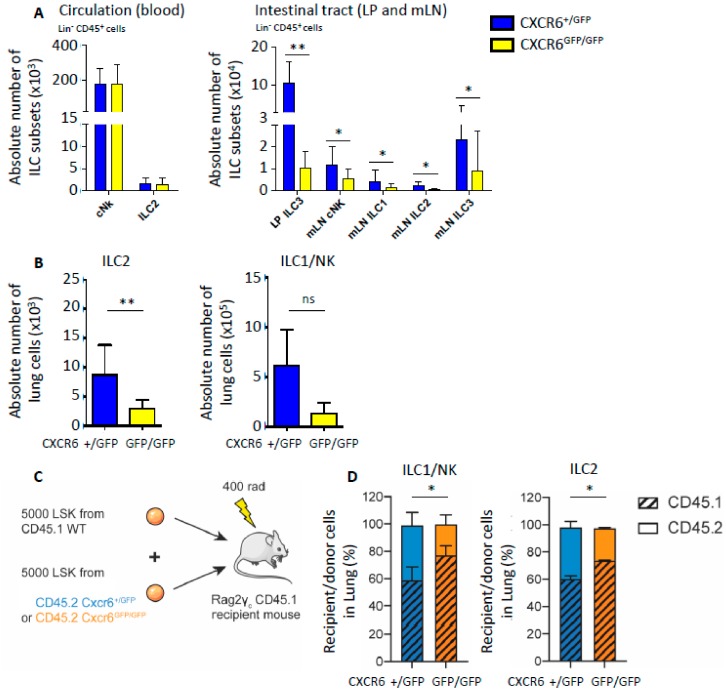
Diverse effects of CXCR6 deficiency on ILC tissue distribution. (**A**) Absolute numbers of different ILC subsets in CD45^+^ Lin^−^ (CD3, CD5, CD19, TCRβ, TCRγδ, Gr1, Ter119, CD8α and F4/80) blood / Lamina Propria (LP) / mesenteric Lymph Node (mLN) cells from Rag^−/−^ CXCR6^+/GFP^ and Rag^−/−^ CXCR6^GFP/GFP^ mice. (+/GFP = 7, GFP/GFP = 7) (**B**) Absolute number of ILC2 (CD45^+^ Lin^−^ NK1.1^−^ Sca1^+^) and ILC1/NK (CD45^+^ Lin^−^ NK1.1^+^ NKp46^+^) lung cells from Rag^−/−^ CXCR6^+/GFP^ and Rag^−/−^ CXCR6^GFP/GFP^ mice. (+/GFP = 6, GFP/GFP = 4) (**C**) Protocol of competitive experiment, 5000 LSK cells from CD45.1 C57Bl/6J mice and 5000 LSK cells from CD45.2 Rag^−/−^ CXCR6^+/GFP^ or Rag^−/−^ CXCR6^GFP/GFP^ mice were injected in irradiated (400 rad) Rag^−/−^ γc^−/−^ CD45.1. (**D**) Repartition of CD45.1 and CD45.2 CXCR6^+/GFP^ or CXCR6^GFP/GFP^ reconstituted cells in ILC1/NK and ILC2. (+/GFP = 4, GFP/GFP = 4) (* *p* < 0.05 or ** *p* < 0.01)

**Figure 3 ijms-20-05493-f003:**
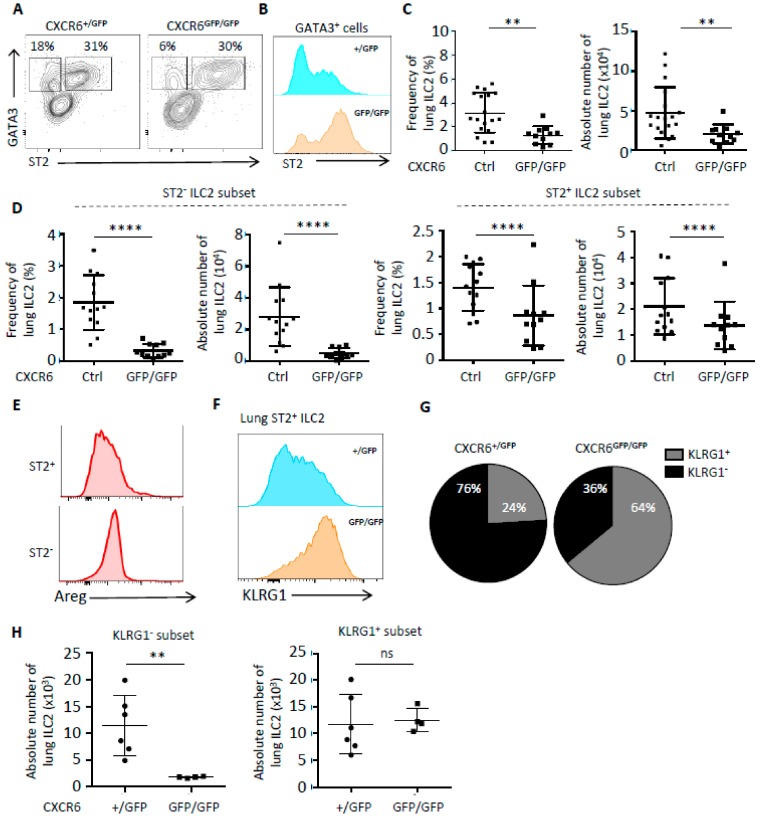
CXCR6 deficiency modifies the phenotype and frequency of ILC2 in an inflammatory context (papain stimulation) (**A**) Flow cytometry analysis of CD45^+^ Lin^−^ (CD3, CD5, CD19, TCRβ, TCRγδ, Gr1, Ter119, CD8α and F4/80) lung cells from Rag^−/−^ CXCR6^+/GFP^ and Rag^−/−^ CXCR6^GFP/GFP^ mice. Different populations of ILC2 are obtained depending on the ST2 and GATA3 expression. (+/GFP = 13, GFP/GFP = 11) (**B**) ST2 MFI of GATA3^+^ lung cells from Rag^−/−^ CXCR6^+/GFP^ and Rag^−/−^ CXCR6^GFP/GFP^ mice. (+/+ = 5, +/GFP = 13, GFP/GFP = 11) (**C**) Frequency and absolute number of lung ILC2 (CD45^+^ Lin^−^ GATA3^+^) from Rag^−/−^ CXCR6^+/+^ and Rag^−/−^ CXCR6^+/GFP^ (Ctrl), and Rag^−/−^ CXCR6^GFP/GFP^ mice. (Ctrl = 18, GFP/GFP = 11) (**D**) Frequency and absolute number of lung ST2^−^ ILC2 and ST2^+^ ILC2 (CD45^+^ Lin^−^ GATA3^+^) from Rag^−/−^ CXCR6^+/+^ and Rag^−/−^ CXCR6^+/GFP^ (Ctrl), and Rag^−/−^ CXCR6^GFP/GFP^ mice. (Ctrl = 18, GFP/GFP = 11) (**E**) Flow cytometry analyses of lungST2^+^ ILC2 (CD45^+^ CD3^−^, NK1.1^−^, Sca1^+^, CD127^+^) and lung ST2^−^ ILC2 (CD45^+^ CD3^−^, NK1.1^−^, Sca1^+^, CD127^+^) from C57Bl6/J mice for Amphiregulin (Areg) secretion. (*n* = 4) (**F**) KLRG1 MFI of GATA3^+^ ST2^+^ lung cells from Rag^−/−^ CXCR6^+/GFP^ and Rag^−/−^ CXCR6^GFP/GFP^ mice. (+/GFP = 6, GFP/GFP = 4) (**G**) KLRG1^+^ and KLRG1^−^ repartition in GATA3^+^ ST2^+^ lung cells from Rag^−/−^ CXCR6^+/GFP^ and Rag^−/−^ CXCR6^GFP/GFP^ mice. (+/GFP = 6, GFP/GFP = 4) (**H**) Frequency and absolute number of KLRG1^+^ ST2^+^ and KLRG1^−^ ST2^+^ lung ILC2 (CD45^+^ Lin^−^ GATA3^+^) from Rag^−/−^ CXCR6^+/GFP^ and Rag^−/−^ CXCR6^GFP/GFP^ mice. (+/GFP = 6, GFP/GFP = 4) (** *p* < 0.01 or **** *p* < 0.001).

**Figure 4 ijms-20-05493-f004:**
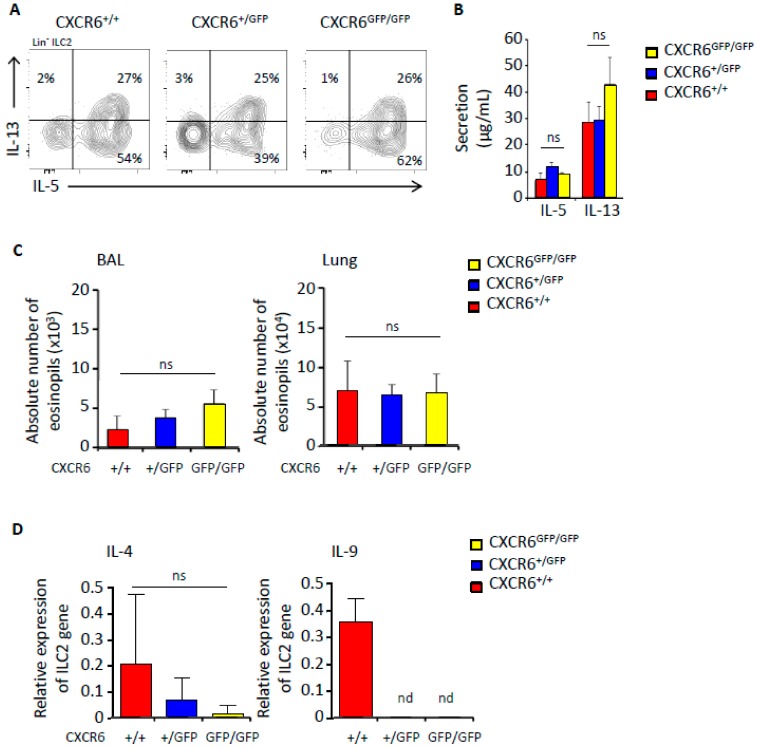
Maintenance of type 2 secretion by ILC2 in CXCR6 deficient mice after papain induction. (**A**) Flow cytometry analysis of CD45^+^ Lin^−^ (CD3, CD5, CD19, TCRβ, TCRγδ, Gr1, Ter119, CD8α and F4/80) NKp46^−^ Sca1^+^ ICOS^+^ lung cells from Rag^−/−^ CXCR6^+/+^, Rag^−/−^ CXCR6^+/GFP^ and Rag^−/−^ CXCR6^GFP/GFP^ mice. Different subsets are disigned in function of IL-5 and IL-13. (+/+ = 5, +/GFP = 13, GFP/GFP = 11) (**B**) Dosage by ELISA of IL-5 and IL-13 secretions from BAL of Rag^−/−^ CXCR6^+/+^, Rag^−/−^ CXCR6^+/GFP^ and Rag^−/−^ CXCR6^GFP/GFP^ mice. (+/+ = 5, +/GFP = 13, GFP/GFP = 11) (**C**) Absolute number of eosinophils (CD45^+^ Lin^−^ SiglecF^+^ CD11b^+^) from Rag^−/−^ CXCR6^+/+^, Rag^−/−^ CXCR6^+/GFP^ and Rag^−/−^ CXCR6^GFP/GFP^ mice in BAL (left graph) and lung (right graph). (+/+ = 5, +/GFP = 13, GFP/GFP = 11) (**D**) 1000 ILC2 (Lin^−^ CD45^+^ NK1.1^−^ ST2^+^) were sorted from Rag^−/−^ CXCR6^+/+^, Rag^−/−^ CXCR6^+/GFP^ and Rag^−/−^ CXCR6^GFP/GFP^ mice. After mRNA extraction, gene study was performed by real time RT-PCR. Relative expression of IL-4 and IL-9 genes (expression compared to Hprt gene), (+/+ = 6, +/GFP = 6, GFP/GFP = 6).

**Figure 5 ijms-20-05493-f005:**
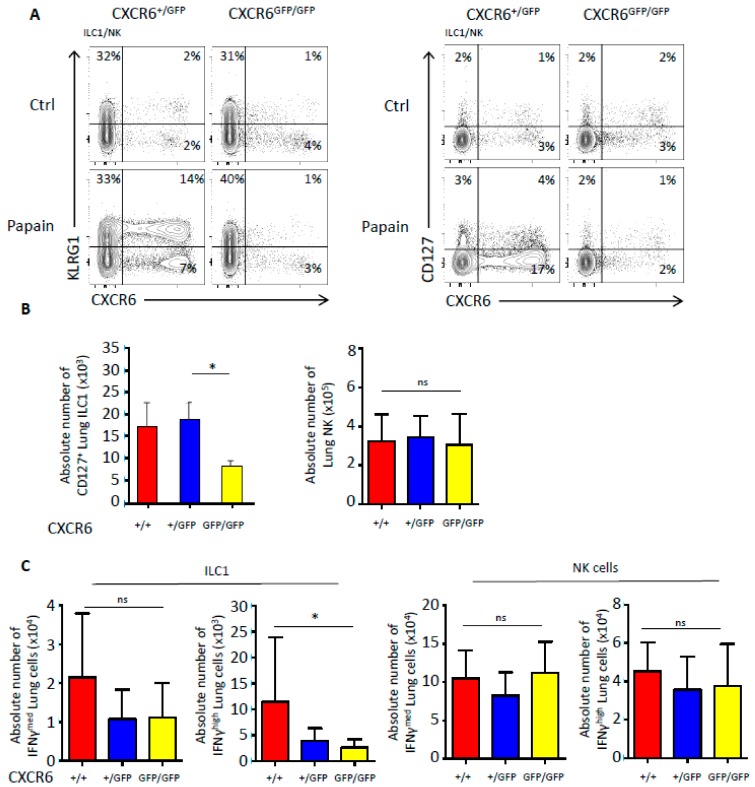
CXCR6 deficiency alters ILC1 numbers and functions after papain induction. (**A**) Flow cytometry analyses of ILC1/NK (Lin^−^:CD3, CD5, CD19, TCRβ, TCRγδ, Gr1, Ter119, CD8α, F4/80; CD45^+^ NK1.1^+^ NKp46^+^) lung cells from Rag^−/−^ CXCR6^+/GFP^ and Rag^−/−^ CXCR6^GFP/GFP^ mice. Different subsets are distinguished by the expression of KLRG1 and CXCR6 (left panel of graphs) or CD127 and CXCR6 (right panel of graphs). (**B**) Absolute numbers of NK cells (Lin^−^ CD45^+^ NKp46^+^ EOMES^+^) and ILC1 (lin^−^ CD45^+^ NKp46^+^ CD127^+^) from Rag^−/−^ CXCR6^+/+^, Rag^−/−^ CXCR6^+/GFP^ and Rag^−/−^ CXCR6^GFP/GFP^ mice. (**C**) Absolute number of IFNγ^med^ and IFNγ^high^ NK cells (Lin^−^ CD45^+^ NKp46^+^ CD127^−^) (2 left graphs) and ILC1 (lin^−^ CD45^+^ NKp46^+^ CD127^−^) (2 right graphs) from Rag^−/−^ CXCR6^+/+^, Rag^−/−^ CXCR6^+/GFP^ and Rag^−/−^ CXCR6^GFP/GFP^ mice. (+/+=10, +/GFP=9, GFP/GFP=8) (**p* < 0.05).

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
