# Peer review of "Maintenance of Type 2 Response by CXCR6-Deficient ILC2 in Papain-Induced Lung Inflammation"

_ijms, 2019, doi:10.3390/ijms20215493_

Round 1
Reviewer 1 Report
In this manuscript, Meunier and colleagues address the role of CXCR6 in NK cells and ILC2s during papain induced lung inflammation using CXCR6-GFP KI/KO mice on a RAG2-/- background. They find:
1) CXCR6 KO reduces ILCs by 3 fold.
2) ILC2 from CXCR6KOs express more ST2 and more KLRG1 but are less frequent in the lungs.
3) Despite their lower frequency, the cells are equally capable of producing IL-5 and 13.
4) Fewer IFNg secretting ILC1 accumulate in the lung after inflammation but that NK cells were similar.
They conclude that CXCR6 plays an important role in both homeostatic and inflammatory homing of ILCs and NK cells to the lungs.
I have two main queries related to this manuscript but the main one concerns the use of RAG2 KO mice as the background strain for all analyses in this study. While I understand that this offers the advantage of increased ILC and NKs for analyses (ie...these rare cells are enriched in the absence of T and B cells), it also adds a confounding factor: How much of the observed defects are due to the absence of T and B cells? Would the observed change in cell frequencies or marker expression occur in a WT background or are they selectively enhanced due to the loss of normal resident lymphoid populations. To address this question it would be important to show similar trends exist in CXCR6 KO animals on a WT background (not a repeat of all analyses, just enough comparisons to confirm that these changes are intrinsic rather than related to loss of gd or ab T cells in the lung for example.)
Specific queries:
1) do all ILC2s express CXCR6 and are most NK cells negative in WT mice or only in RAG mice?
2) For the competitive reconstitution experiments, the authors conclude CXCR6 is important for ILC engraftment of lung. Yet in the introduction they argue that it is also important for bone marrow retention of ILC precursors. What then is the basis to conclude that CXCR6 is key for lung homing in the competitive transplants? If the KLS cells transplanted into irradiated recipients require CXCR6 for bone marrow homing, retention, and maturation in this compartment, then it is equally possible that the effects they observe are due to an upstream lesion in ILC progenitor maturation? This could be further compounded by the fact that the authors only allow 4 weeks for reconstitution rather than the more traditional 10-20 weeks for HSC transplantation.
3) iILC2s have also, by some authors, been called "memory ILC2s" implying that they come orginally from ST2+ ILC2s which have previously been activated, change their circulation patterns and produce more cytokine more quickly in response to a second inflammatory insult. Do the CXCR6 KO experiments allow the authors to conclude whether these are indeed "memory cells" that have previously been activated or a completely different ILC2 lineage.
Author Response
I have two main queries related to this manuscript but the main one concerns the use of RAG2 KO mice as the background strain for all analyses in this study. While I understand that this offers the advantage of increased ILC and NKs for analyses (ie...these rare cells are enriched in the absence of T and B cells), it also adds a confounding factor: How much of the observed defects are due to the absence of T and B cells? Would the observed change in cell frequencies or marker expression occur in a WT background or are they selectively enhanced due to the loss of normal resident lymphoid populations. To address this question it would be important to show similar trends exist in CXCR6 KO animals on a WT background (not a repeat of all analyses, just enough comparisons to confirm that these changes are intrinsic rather than related to loss of gd or ab T cells in the lung for example.)
We agree with Reviewer 1 that ILC are perturbed in RAG KO background as the literature already showed that the absence of T and B cells have consequences, especially in terms of ILC absolute numbers and increased activated phenotype. Even if we considered that comparing the absence (GFP/GFP) versus the presence (GFP/+) of CXCR6 expression in similar RAGKO background is fair since both conditions to be compared are equally missing T and B cells, we decided to implement the information on CXCR6 in WT background. We did supplementary experiments using CXCR6GFP/+ and CXCR6GFP/GFP mice on a WT background to question whether the absence of T/B cells is important on the CXCR6 role. We thank reviewer 1 for that question that increases the message of our ms on the CXCR6 role and target its effect to iILC2 subset mostly. We documented the CXCR6 expression by pulmonary ILC2 and NK cells in steady state but also after papain challenge. These results are shown in supplementary figure 2, 3, 4.Specific queries:do all ILC2s express CXCR6 and are most NK cells negative in WT mice or only in RAG mice?
We used 3 CXCR6GFP/+ mice on a WT background to obtain data on CXCR6 expression by ILC2 KLRG1-, ILC2 KLRG1+ and ILC1/NK cells and produce a supplementary figure 2 for the ms. We indicated the frequency of CXCR6 subsets (as a mean of the percentage obtained from the 3 individual mice). The ILC2 KLRG1+ subset is mostly CXCR6+ in both rag or WT background (74% and 85%) whereas the ILC2 KLRG1- subset is less prone to express CXCR6 in a WT background (13% compared to 44%). The pulmonary NK/ILC1 population is still poorly expressing CXCR6 in a WT background with only 8% of the cells being CXCR6+. Hence, we concluded that the main difference found is that CXCR6 is less frequently expressed by resident lung KLRG1- ILC2 in case of RAG deficiency.
For the competitive reconstitution experiments, the authors conclude CXCR6 is important for ILC engraftment of lung. Yet in the introduction they argue that it is also important for bone marrow retention of ILC precursors. What then is the basis to conclude that CXCR6 is key for lung homing in the competitive transplants? If the KLS cells transplanted into irradiated recipients require CXCR6 for bone marrow homing, retention, and maturation in this compartment, then it is equally possible that the effects they observe are due to an upstream lesion in ILC progenitor maturation? This could be further compounded by the fact that the authors only allow 4 weeks for reconstitution rather than the more traditional 10-20 weeks for HSC transplantation.
The reviewer is right, we should increase our conclusion on the possible role of CXCR6 after reconstitution. CXCR6 is probably not limited to a role in lung ILC engraftment and we have to consider that BM progenitors are also perturbed with possibly less progenitors leaving the BM. Moreover, the 2 processes (engraftment + progenitor retention) might occur and are not mutually exclusive. Hence, we changed the ms accordingly in the text. Moreover, we thank the reviewer for having pointed out the discrepancy on the duration of reconstitution. Our experiments are performed 8 weeks after HSC reconstitution, as soon as NK cells could be detected in the blood of the mice. This was also accordingly corrected in the ms.iILC2s have also, by some authors, been called "memory ILC2s" implying that they come originally from ST2+ ILC2s which have previously been activated, change their circulation patterns and produce more cytokine more quickly in response to a second inflammatory insult. Do the CXCR6 KO experiments allow the authors to conclude whether these are indeed "memory cells" that have previously been activated or a completely different ILC2 lineage.
Lung iILC2 have been demonstrated by Huang et al. to be a pool of cells derived from the intestinal lamina propria that is different from the resident lung nILC2. We added a supplementary figure (S5) to provide the percentage of ILC2 in the intestine that are CXCR6+ and could then migrate to the lung as the iILC2 pool. These cells are decreased in numbers in mLN of CXCR6KO (our figure 2A) explaining the global decrease of ST2- ILC2 in the lung of CXCR6KO mice after papain challenge. In conclusion, pulmonary KLRG1+CXCR6+ ILC2 are probably coming from the intestinal lamina propria. We also thought about the possibility that CXCR6 could be a marker of “memory” ILC2 as suggested by reviewer 1. Since, we did not perform “real” secondary immunizations to prove this statement, we did not propose it in the first version of the ms. However, we agree that it should be discussed as it is a very interesting point. Moreover, our new exp on the WT background in which ILC2 are not activated by default tend to confirm that CXCR6 is less frequently expressed by lung nILC2 in absence of T/B cells. Our supplementary experiments of papain challenged mice in a WT background is mostly showing identical results than those provided in a RAG deficient background. Indeed, even if ILC were reduced in numbers, iILC2 were decreased in numbers and KLRG1+ cells still show higher levels of KLRG1 in absence of CXCR6. Accumulation of KLRG1+ cells in the WT background is less evident than in the RAG background since in the latter case T cells are not entering in competition with ILC2 for space (especially in the lamina propria). In the WT background, CXCR6 deficiency is more stringent and deeply affects the number of circulating CXCR6+KLRG1+ iILC2 from the intestine to the lung. We suspect that the presence of T cells is decreasing the probability of iILC2 to compete for reaching their niches and proper cytokines for their expansion. CXCR6 deficient mice in a WT background were also analyzed for the repartition of ILC1/NK populations after papain stimulation (figure S4). Globally, the absence of CXCR6 in mice with T and B cells had similar issues with a stronger decrease of lung ILC1 numbers. It is interesting to observe that in presence of T and B cells, NK cells are also importantly decreased after inflammation while their frequency of CXCR6+ cells are unchanged at a low 8% rate. It appears that the recruitment of NK cells in case of lung inflammation might be indirectly dependent on CXCR6 expression by lymphocytes.
Reviewer 2 Report
major:
sample preparation of hematopoietic cells from lungs is tricky. Usually, leukocytes are localized in lung vessels or in the interstitium / intraalveolar. It is not described how the authors excluded contamination of the "lung" samples with cells from peripheral blood (e.g. previous injection of CD45 Ab to label intravascular cells). Please comment on how cells were isolated and/OR if "bulk" lung cells were used provide comparative data to blood ILCs. Figure 2, 4, and 5: Please provide information on data presentation (e.g. SEM, mean, median including error bars) or better provide scattered plot as in Figure 3. SEM is not stand of the art. Figure 2: What is the overall reconstitution of cells obtained from CXCR6+ vs. CXCR6- LSKs? Is there a general defect for CXCR6- cells to home into and repopulate the bone marrow.
minor:
could the authors speculate on why KLRG1+ cells accumulate in the lungs of CXCR6- mice? Are these the cytokine producers in CXCR6+ animals? Please discuss which other chemokine receptors are involved in homing of lymphoid cells to the inflamed lung?
Author Response
Reviewer 2
Sample preparation of hematopoietic cells from lungs is tricky. Usually, leukocytes are localized in lung vessels or in the interstitium / intraalveolar. It is not described how the authors excluded contamination of the "lung" samples with cells from peripheral blood (e.g. previous injection of CD45 Ab to label intravascular cells). Please comment on how cells were isolated and/OR if "bulk" lung cells were used provide comparative data to blood ILCs.
We thank Reviewer 2 to remind us to show this important point. We previously did series of experiments to observe whether our exsanguination protocol could eliminate circulating ILC2 as efficiently as an ivCD45 antibody staining (Figure S1). All our lung preparations along the ms were done using exsanguination that remove most circulating ILC2 as shown in supplementary figure 1 using control intravital injections of ivCD45 antibody (for two mins) before the sacrifice of the mice. The figure proves that most of the circulating ILC2 are gone after exsanguination with only 4% of remaining intravascular ILC2. On the contrary, NK cells that are known to be all intravascularly positioned remained after exsanguination but with a clear positive ivCD45 staining.
Figure 2, 4, and 5: Please provide information on data presentation (e.g. SEM, mean, median including error bars) or better provide scattered plot as in Figure 3. SEM is not stand of the art.
There is no SEM in our figures. All figures were designed using mean including error bars and we added this statement in the mat and meth part of this new version.
Figure 2: What is the overall reconstitution of cells obtained from CXCR6+ vs. CXCR6- LSKs? Is there a general defect for CXCR6- cells to home into and repopulate the bone marrow.
The bone marrow ILC precursors were not analyzed in these mice after reconstitution. However, as expected there was no difference for the LSK compartment between CXCR6 sufficient versus deficient mice consistent with the fact that CXCR6 expression is only starting at the a-lymphoid progenitor stage. A sentence has been added in the ms.
minor:
could the authors speculate on why KLRG1+ cells accumulate in the lungs of CXCR6- mice?
Lung iILC2 have been demonstrated by Huang et al. to be a pool of cells derived from the intestinal lamina propria that is different from the resident lung nILC2. We added a supplementary figure (S5) to provide the percentage of ILC2 in the intestine that are CXCR6+ and could then migrate to the lung as the iILC2 pool. These cells are decreased in numbers in mLN of CXCR6KO (our figure 2A) explaining the global decrease of ST2- ILC2 in the lung of CXCR6KO mice after papain challenge. In conclusion, pulmonary KLRG1+CXCR6+ cells are probably coming from the intestinal lamina propria. We consider that their accumulation in the CXCR6 deficient mice is due to an inability of ILC2 to be correctly localized in the lung to further differentiate into ST2+KLRG1+ and then ST2- nILC2.
Are these the cytokine producers in CXCR6+ animals?
Reviewer 2 is right, we added in the discussion the following statement: It was already shown that in case of IL25 or helminth infections, iILC2 are strong producer of cytokines. Since CXCR6+ cells are mostly pulmonary iILC2, they could be attributed to the pool of cytokine producers. However, we do not know whether this is directly or indirectly linked to CXCR6 expression.
Please discuss which other chemokine receptors are involved in homing of lymphoid cells to the inflamed lung?
We added these sentences as CXCR6 is probably indirectly acting on ILC2 circulation:
CXCR6 may have an indirect role in the homing of ILC2 to the lungs and others chemokines may be important for the homing to the lungs as reported by others. CCR4, CCR8 and CCR1 are expressed by ILC2 in steady-state conditions and are implicated in lung ILC homeostasis with a decrease of ILC2 numbers in case of CCR4 deficiency (Knipfer JEM 2019, Puttur sciences Immunol 2019). After IL33 induced inflammation, CCR8 and CCR1 are increased at the surface of ILC2 and blockade of CCR8 could reduce both ILC2 motility and secretion suggesting an important role for their homing to the lungs in inflammatory context (Puttur sciences Immunol 2019). Moreover, the expression of CCR8 has been correlated with the expression of its CCL8 ligand by inflamed epithelial cells (Puttur sciences Immunol 2019). After the induction of inflammation, lungs ILC2 also express CCL1, another CCR8 ligand, and this expression enhance their motility, proliferation and survival when interacting with the inflamed epithelium (Knipfer JEM 2019) .
Round 2
Reviewer 1 Report
The authors have addressed each of my concerns